# Impact of Accessory Corpus Luteum Induced by Gonadotropin-Releasing Hormone or Human Chorionic Gonadotropin on Pregnancy Rates of Dairy Cattle following Embryo Transfer: A META-Analysis

**DOI:** 10.3390/vetsci10050309

**Published:** 2023-04-23

**Authors:** Fang Chen, Yi’an Hou, Xiaoqing Zhu, Cheng Mei, Rihong Guo, Zhendan Shi

**Affiliations:** 1Institute of Animal Science, Jiangsu Academy of Agricultural Sciences, Nanjing 210014, China; 2Science and Technology Industry Development Center, Chongqing Medical and Pharmaceutical College, Chongqing 401331, China; 3Dongying Austasia Modern Dairy Farm Co., Ltd., Dongying 257345, China; 4Key Laboratory of Crop and Animal Integrated Farming, Ministry of Agriculture, Nanjing 210014, China; 5Jiangsu Key Laboratory for Food Quality and Safety-State Key Laboratory Cultivation Base of Ministry of Science and Technology, Jiangsu Academy of Agricultural Sciences, Nanjing 210014, China

**Keywords:** cow, pregnancy, embryo transfer, accessory corpus luteum

## Abstract

**Simple Summary:**

In recent decades, embryo transfer in dairy cows has been widely applied, and improving the conception rate after embryo transfer has become a major challenge in dairy cow reproduction. Progesterone is a critical hormone during early pregnancy in the cow. One strategy to increase progesterone levels and improve the fertility of dairy cows is inducing accessory corpus luteum through the administration of GnRH or human chorionic gonadotropin after embryo transfer. Many studies tried to improve conception rate of ET recipients, but failed to deliver a clear consensus on the therapeutic benefits of such treatments. Thus, a meta-analysis to evaluate the efficacy of hCG and GnRH in improving pregnancy rates after ET was carried out in this review. Our results indicated that the use of GnRH or hCG can improve pregnancy rates in dairy recipient cows with very poor fertility (<40%), but does not have a significant effect on cows with good fertility. Compared with GnRH or GnRH analogue, hCG treatment acquired better improvement in conception rates after ET. Treatment between days 5 and 7 after synchronized ovulation was beneficial, while later than this was not. Furthermore, the treatment was more effective in parous lactating dairy cows compared with heifers. This review may have great significance in the management of reproductive performance on modern dairy farms.

**Abstract:**

The circulation of progesterone (P4) concentrations of recipients has positive correlations with embryo survival and pregnancy success of embryo transfer (ET) in dairy cows. One strategy to improve P4 concentration is the administration of gonadotropin-releasing hormone (GnRH) or human chorionic gonadotropin (hCG), thereby inducing the formation of accessory corpus luteum (CL). This study aimed at determining the efficacy of GnRH or hCG treatment regarding embryo transfer (ET) and providing a better clinical veterinary practice guidance. A meta-analysis was conducted on the data from 2048 treated recipient cows and 1546 untreated cows. By inducing the formation of accessory CL with GnRH (100 µg), GnRH analogue Buserelin (8–10 µg), or hCG (≥1500 IU) 5–11 days after synchronized ovulation, hCG alone achieved an improvement (RR = 1.39, *p* < 0.05), while GnRH and GnRH analogue did not result in significant changes (RR = 1.04, *p* = 0.26). Treatment with GnRH or hCG 5–7 days after synchronized ovulation was associated with increased chances of pregnancy compared with later treatment (11–14 days). Owing to the treatment, the pregnancy rate of cows with very poor fertility (<40%) was improved, while that of cows with good fertility (≥40%) was not affected. Treatment with GnRH or hCG greatly improved pregnancy rates of parous lactating cows (RR = 1.32, *p* < 0.05) compared with heifers (RR = 1.02, *p* > 0.05). Additionally, as indicated by pregnancy loss analysis, the treatment had no benefit on late embryo/early fetus survival at days 28–81. In conclusion, the induction of accessory CL with GnRH or hCG may benefit fertility and have important implications for the management of reproductive performance in the dairy industry.

## 1. Introduction

In recent years, with the increase in genetic merit for milk production worldwide, the reproductive efficiency of dairy cows has declined [1,2], and it further descended by 10% in summer due to heat stress [3]. The decrease in pregnancy rates might be caused by insufficient follicle or oocyte development, which is very sensitive to environmental and physiological stresses [4,5]. To avoid the adverse effect of heat stress on oocyte development and maturation, ovulation, and early embryonic development, embryo transfer (ET) was widely adopted to facilitate fertility in dairy cows as a reproductive technology [6,7,8,9]. ET, coupled with many new biotechnics, including gene editing and somatic cell nuclear cloning, becomes a tool for genetic selection [10] and breeding [11]. Fresh or thawed blastocysts are usually transferred to recipients about 7 days after anticipated ovulation [12]. Although some groups failed to find any difference in conception rates between dairy cows of timed artificial insemination (TAI) or ET, others observed enhanced fertility with ET contrasted with TAI, because ET selects embryos based on development potential and can bypass the most critical thermosensitive periods of the oocyte or early cleavage-stage embryo [13]. Hence, by employing ET technology, the challenges caused by failure of oocyte maturation, poor embryo quality, and low tolerance to heat can be minimized, thus theoretically promoting the pregnancy rate of ET recipients compared with that of inseminated cows [14].

Early embryo mortality before implantation is another major factor that reduces pregnancy rate [15,16]. Up to 40–60% of embryos die within the first two weeks after fertilization [17,18,19,20], and the embryo loss is more severe in high-yielding cows [19,21]. Following ET, the blastocyst develops into an ovoid and then tubular form after hatching from the zona pellucida. The embryo begins to elongate on days 12–14, forming a filamentous conceptus [16]. A significant proportion of losses is likely associated with defects in the process of conceptus elongation, during which the embryo secretes interferon tau to prevent luteolysis [22]. In particular, successful pregnancies of dairy cows need sufficient progesterone, thereby stimulating endometrial secretion production. As discovered by studies, P4 secreted by CL is important throughout the pregnancy of cows [23]. Early conception failure and embryo mortality can be caused by sustained low P4 circulation between 5 and 10 days following ovulation and fertilization [24]. Thereof, endometrial secretion is conducive to embryo development and implantation [25,26]. Insufficient progesterone circulation levels are associated with unfavorable conceptus development [27], and it cannot restrain luteolytic signal development in dairy cows [28,29]. Moreover, compared with heifers, lactating dairy cows exhibited a decrease in plasma P4 levels, probably due to the increase in liver blood flow and active P4 metabolism with rising milk production [30]. Hence, increasing plasma P4 concentration could improve fertility, exclusively for lactating dairy cows.

In view of the consensus on the significance of progesterone, there are numerous studies on post-insemination progesterone supplementation [31,32,33]. A common strategy to improve P4 levels after AI or ET is inducement of ovulation and formation of accessory corpus luteum (CL) with gonadotropin-releasing hormone (GnRH), GnRH agonists, or human chorionic gonadotropin (hCG) when the dominant follicle (DF) of the first follicular wave is present on the ovary. Accessory CL induced by hCG during diestrus not only altered follicular and luteal dynamics, but also deferred and prolonged the luteolytic process [34]. There are several other studies [32,35,36] that give systemic reviews of the impact of progesterone (P4) supplementation or hCG and GnRH administration on the fertility of dairy cows after artificial insemination (AI), indicating an overall improvement in pregnancy rate. As reported by Yan [35], the progesterone supplementation given in the 3rd–7th days was beneficial, while that given earlier or later than this time period was not beneficial. Others [36] discovered that treatment with GnRH 10 days after AI was also conducive to pregnancy after AI. However, the effects of inducing an accessory CL in ET programs have not been well systemically analyzed.

Many studies tried to increase P4 concentration to improve the fertility of ET cows, but these studies failed to deliver a clear consensus on the therapeutic benefits of such treatments. Some reported a beneficial effect of GnRH or hCG on conception after ET, while others indicated no obvious improvement. Thus, the objective of this study was to evaluate the efficacy of hCG and GnRH in the improvement of pregnancy rates of dairy cows in the luteal phase by performing a meta-analysis of the data available from all studies on hCG and GnRH treatment.

## 2. Materials and Methods

### 2.1. Identification and Inclusion Criteria of Studies

In this study, PubMed and Web of Science databases were selected to identify eligible studies of recipients with post-ovulation treatment. Keywords used in MeSH search were limited to “Title/Abstract”, including (i) embryo transfer or ET, (ii) dairy cattle or cow * or heifer *, (iii) progesterone or GnRH or hCG, and (iv) reproductive or pregnancy or conception. The databases were searched by two of our investigators independently, with each article reviewed as well. For this meta-analysis, only the studies published in English were included.

### 2.2. Selection Criteria of Studies

Based on Preferred Reporting Items for Systematic Reviews and Meta-Analysis (PRISMA) guidelines, the principle of PICO (Problem/Patient/Population, Intervention, Comparison, and Outcome) needed to be followed. Inclusion criteria were listed, including (i) population, namely recipient dairy cows in timed ET; (ii) intervention, namely administration of GnRH or hCG; (iii) comparison with control groups not given treatment with any progesterone concentration; and (iv) outcome, namely conception status diagnosis from days 25 to 100 after estrus. Repetitive articles were eliminated, followed by screening, thereby confirming if the resulting quantity of articles were sufficient for meta-analysis (Figure 1).

### 2.3. Data Extraction and Summary of Experimental Animals

The relevant data parameters were extracted by two independent investigators (F.C. and X.Z.), with disagreements solved by discussion with section partner. Finally, the data parameters below were extracted, namely study ID, first author, publication year, region, features of animal model (species, BCS, and age), information of treated/control (untreated) group (including treatment type (GnRH/hCG), number of animals, dosage, and administration time), and outcome indicators (including pregnancy status and diagnosis time).

### 2.4. Classification of Studies for Analysis

In view of this meta-analysis, four subgroup analyses were carried out according to the type of treatment, time of treatment, parity of recipients, and the initial pregnancy rate. The impact of treatment on pregnancy loss rate was determined as well.

Regarding the first analysis, classification was applied to the studies based on administration type, namely GnRH (GnRH or GnRH analogue) or hCG.

In the second analysis, classification was applied to the treatment time, namely days 5–7 or days 11–14 of estrus cycle, corresponding to ovulation of dominant follicle (DL) of first or second follicular wave, respectively.

As for the third analysis, subgroups were divided according to parity, namely nulliparous or parous.

As for the fourth analysis, classification was given to the studies based on initial pregnancy rate, namely poor (<40%), and good (≥40%). The initial pregnancy rate was defined as pregnancy rate or conception rate of untreated recipients in each study.

In the fifth analysis, only studies that reported pregnancy diagnoses before day 35 and after day 42 were included; the pregnancy loss rates were compared between treatment and control groups.

### 2.5. Statistical Analysis

RevMan 5.4.1 was used in the meta-analyses. Heterogeneity degree was evaluated with Cochran Q statistic and *I*^2^ statistic of the proportion of total variation regarding heterogeneity. A value *I*^2^ greater than 50% was considered to display the higher heterogeneity, with random-effect model applied. Otherwise, fixed-effect model needed to be adopted. Identification was given to heterogeneity source, and subgroup analysis was conducted. *p* < 0.05 was considered statistically significant. Risk ration (RR) and 95% confidence intervals (CIs) of each analysis were exhibited by creating forest plots and in graphic manner. A black box and horizontal line were used in representing each study, and they corresponded to the point estimated 95% CI of each study. Contribution of the study to overall effect was represented by the size of box in forest plot. Pooled RR was estimated, and the diamonds at bottom of the same plot were used in representing 95% CI. Outliers were identified by conducting an influential case diagnosis.

To determine if there was possible interaction and bias between treatment type, time, dosage, and parity of recipients, correlation test was carried out with GraphPad Prism 9.0.0. *p* < 0.05 was considered statistically significant, namely, the two variables were correlated to each other. No significant interaction or correlation was detected among treatment type, time, dosage, and parity of recipients (Appendix A).

## 3. Results

### 3.1. Literature Screening

A total of 277 studies with potential relevance were identified by searching PubMed, Science Direct, and Web of Science. Afterwards, 122 repetitive publications were excluded, 136 publications were excluded by considering inclusion or exclusion criteria, and one publication was eliminated for no controls. As a result, 9 eligible publications and 12 independent studies were included for this study (shown in Figure 1).

### 3.2. Characteristics of Studies Included

The 12 studies included exhibited difference in parity (multiparity/primiparity), treatment type, and pregnancy rate, etc. General characteristics of the studies included are displayed in Table 1. In view of parity, five studies used parous lactating cows as recipients, six studies adopted heifers, and one study employed both primiparous and multiparous cows. Regarding the treatment, GnRH was adopted in seven studies and hCG in five studies. For the 12 studies, the controls presented a great difference in the chance of pregnancy rate, and the treatments showed different effects on pregnancy rate (Figure 2). The funnel plot did not suggest a bias of publications.

### 3.3. Effect of Treatment Type

As exhibited by the meta-analysis (Figure 3), the experimental group exhibited a great increase in pregnancy rate with the administration of hCG or GnRH 5 to 11 days after synchronized ovulation (RR = 1.09, *p* < 0.05). Subgroup analyses were further conducted based on the different hormone treatments. As indicated in Figure 1, the pregnancy rate presented no obvious increase when treating cows with 100 µg GnRH or GnRH analogue (8–10 µg Buserelin or 750 µg Deslorelin) (RR = 1.04, *p* = 0.26). However, it showed an obvious increase when treating cows with 1500–3300 IU hCG (RR = 1.39, *p* < 0.05).

### 3.4. Effect of Treatment Time

In view of the 12 available studies, 9 studies treated the animals at ET or two days before ET, inducing ovulation of the dominant follicle (DL) of the first follicular wave. Three studies treated the animals at day 11, inducing DL ovulation of the second follicular wave (Figure 4). Treatment on day 5–7 after synchronized ovulation showed a significant increase (RR = 1.10, *p* < 0.05), while treatment on day 11 reported no obvious change (RR = 1.04, *p* = 0.56).

### 3.5. Effect of Parity

In view of the 12 studies from nine publications, 5 studies were conducted by adopting heifers, 6 studies were conducted by using multiparous cows, and 1 study did not provide any information on parity status. Moreover, subgroup analysis was performed based on the different status of parity, namely primiparous and multiparous. As shown in Figure 5, there was an obvious difference in experimental group and control group in multiparous cows (RR = 1.32, *p* < 0.05). No significant improvement was reported in heifers (*p* = 0.59).

### 3.6. Effect of Initial Pregnancy Rate

A total of 10 studies reported poor initial pregnancy rates (Figure 6) (<40%). Meta-analysis indicated a significant improvement in the pregnancy rates of hCG- and GnRH-treated recipients (RR = 1.14, *p* < 0.05). Two included studies reported a good initial pregnancy rate (>40%), and the treatment showed no positive effect on conception rates (RR = 1.02, *p* = 0.65).

### 3.7. Pregnancy Loss

As for seven studies included, the pregnancy status of cows was checked more than once, thereby evaluating the impact of P4 supplementation on pregnancy loss. As exhibited by meta-analysis (Figure 7), P4 supplementation had no obvious impact on pregnancy loss rates (RR = 0.86, *p* = 0.272).

## 4. Discussion

Current research suggests that hCG treatment improved the conception rate more compared with GnRH or GnRH analog. Treatment 5–7 days after synchronized ovulation was associated with an increased pregnancy rate compared with later treatment. The treatment significantly enhanced the cows’ fertility with very poor fertility (<40%), but had no impact on cows with good fertility (≥40%). Furthermore, the treatment was more effective in parous lactating dairy cows than heifers.

Although some earlier studies found a connection between daily P4 administration and a higher fertility rate in dairy cows [46], there were also opposing results supporting that P4 supplementation after AI or ET was ineffective for fertility in practice [31,32], indicating the treatment type, dosage, and treatment time may impact its efficacy.

Several strategies have been published to increase the circulating P4 during the early luteal phase. These strategies include P4 supplementation during the critical period (days 15–19 after ovulation) [47,48,49] and accessory CL formation induced by GnRH or hCG administration. According to prior research, accessory CL formation increased P4 concentration and reduced estrogen production [50], positively affecting embryonic survival. As shown in this meta-analysis, the induction of accessory CL with hCG presented a higher pregnancy rate after ET than the control treatment. These data supported the notion that P4 supplementation during embryo implantation positively affected dairy cow fertility.

Additionally, the first-wave follicle ovulation induced by GnRH and hCG could change the follicular dynamics during luteolysis [34] and release more time for the embryo to elongate, thereby rescuing some delayed embryos and increasing the pregnancy rate [35,48]. As indicated by subgroup analysis, hCG administration improved consistency, while GnRH administration had various effects on pregnancy rates. Compared with GnRH, hCG binds to LH receptors of ovarian cells independently of the pituitary glands [51], and its effect is longer lasting than that produced by endogenous LH release [52]. Moreover, the CL formed after ovulation induced by GnRH may not be fully functional [53,54]. Consequently, hCG treatment typically increases progesterone levels more than GnRH treatment [55]. The difference within studies of GnRH treatments was likely caused by the different time points of GnRH treatment and the dosage injected.

Notably, earlier studies indicated a change in the overall pregnancy rate based on the time of administration of AI to cows. As reported by Yan [35], progesterone supplementation at the beginning of the luteal phase, namely before day three after insemination, presented no obvious benefit on the pregnancy rate. This had the underlying mechanism that early treatment with progesterone could promote luteolysis development to a time point for sufficient anti-luteolytic response mustered by the embryo. According to the summary of Besbaci [36], treatment after day 10 of AI had a relation with the higher P/AI by comparing it with GnRH treatment before day 10 of AI. Most animals included in this meta-analysis were administrated with GnRH or hCG on the day of ET or two days before, 5–7 days after estrus. Only in three studies were the cows administrated four days after ET, and no obvious reduction or improvement in pregnancy rate was reported, when compared with ET treatment. It was indicated that inducement of accessory CL during the period of the first follicular wave resulted in a significant increase in pregnancy rate. It is feasible to create a more favorable uterine environment for embryonic development by supplementing progesterone during the early phases of embryonic development, thus rescuing some of the delayed embryos and increasing the cows’ fertility. In this study, the ineffectiveness of GnRH treatment on day 11 was consistent with previous results indicating that the ovulatory response to GnRH is poor after day 10 of the cycle due to the lack of a dominant follicle, particularly from day 10 to 14 in two-wave cows, and the inhibitory effect of high progesterone on the induction of an LH surge [56].

Considering that circulating P4 may be limited by the lactating of dairy cows, subgroup analyses was carried out according to parity, namely heifers and parous cows. As demonstrated by the present experiments, GnRH and hCG administration significantly enhanced P/ET in parous cows, showing GnRH and hCG’s utility in lactating cattle’s luteal support.

This study investigated the impact of the initial pregnancy rate, namely the pregnancy rate of untreated recipient cows, on the progesterone supplementation efficacy on pregnancy rate. According to the findings, treatments with GnRH or hCG could improve P/ET in cattle with low initial fertility (<40%), but have no effect on cows with high fertility (>40%). In fixed-time artificial insemination protocol, P4 supplementation improved P/AI in dairy cows with low fertility (<45%), but it did not cause any benefit for cows with relatively good initial fertility [35], as reported. The pregnancy rate in AI cattle may be impacted by irregular ovulation, incorrect estrus, insufficient CL functions, and inadequate circulating P4 for embryo development. Unlike AI cattle, recipient cattle in ET received relatively good embryos, eliminating the potential negative effect of ovulation or oocyte quality. Hence, CL functions became one of the major challenges to fertility and brought about different responses to accessory CL in dairy cows with different initial fertility. The cows with poor initial fertility and insufficient CL showed obvious responses to P4 treatments. In contrast, cows with good initial fertility possibly had sufficient P4. Thus, the effect of further P4 supplementing treatment on embryo implantation was quite minimal.

Given this meta-analysis, by comparing with controls, GnRH and hCG administration had no obvious effect on pregnancy loss rate. Most of the embryo loss occurred in the first 16 days in dairy cattle, and about 5.8% pregnancy loss occurred in days 32–100 [10] in beef cows. Considering the included trials with pregnancy status diagnosed twice, the overall pregnancy loss rates were 7.3% in controls and 7.4% in treatment groups. These results provided solid evidence that embryo survival in the first month after estrus is essential to improving pregnancy rate and economic benefits.

Notably, the current meta-analysis was undertaken by adopting the studies with GnRH or hCG administrated to cows, exploring whether introducing accessory CL could deliver a consensus improvement in pregnancy rate. Besides administering GnRH or hCG, use of equine chorionic gonadotropin (eCG) in synchronization protocols, estradiol/progestin-based protocols, and GnRH-based protocols were the most common strategies to increase pregnancy rates in fixed-time embryo transfer (FTET) [57]. Numerous studies [58,59] indicated that the administration of eCG in synchronization protocol for FTET resulted in increased pregnancy rates, especially in recipients managed under nonoptimal conditions. However, most of the studies used eCG in the synchronization period instead of the early luteal phase. Only one study used eCG to introduce accessory CL, but on beef cows [60]. As for treatment during the early luteal phase, four available studies adopted a controlled internal drug release (CIDR) after ovulation, thereby enhancing P4 concentration and pregnancy rates. Thereof, three studies [47,61,62] treated beef or dairy cows with CIDR containing 1.9 g P4 for 12–13 days, and they all were not designed with enough statistical power for detecting pregnancy differences between control and CIDR groups (28.6 vs. 41.1%, 15.5 vs. 25.0%, and 67 vs. 73%, respectively). According to the other study [63], the pregnancy rates could be reduced by inserting CIDR 4 days before ET (39.7 vs. 21.3% vs. 15.2%, for ET-Control, ET-CIDR-4, and ET-CIDR-14, respectively). These findings were similar to Yan’s meta-analysis conclusion, which concluded that reporting treatment earlier than three days after AI did not correlate with the likelihood of conception per AI.

Nonetheless, the existing studies offer sufficient data for moderator analysis of treatment type and time, parity, and initial fertility. Other factors may impact the pregnancy rate, including the season [64], nutritional status [65], BCS [65], and transferred embryo status (produced in vitro or in vivo, fresh or thawed). For example, the type of embryos (fresh/thawed/semi) and temperature (normal/heat-stressed) may compromise the initial fertility of controls. However, there were insufficient records to examine based on previous research. Climate and management conditions may also impact the fertility of recipients, but we were not able to carry out another subgroups analysis based on insufficient records about climate or season. A further study based on five heat-stressed dairy recipient cow studies indicated that GnRH/hCG improves conception rate in quite high temperatures (Appendix A). The best reproductive management strategies rely on various factors. By undertaking further research, it will be possible to compare the effects of the above moderators on overall reproductive performance, hence directing field application.

## 5. Conclusions

This meta-analysis examined the association between the formation of accessory CLs around ET and ET outcome in dairy cows. The induction of accessory CL with hCG could improve the conception rates of recipient dairy cows, while GnRH and GnRH analogue did not result in significant changes. The results clearly demonstrated that the administration of hCG or GnRH 5–7 days after synchronized ovulation was associated with increased pregnancy rate, compared with later treatment. The induction of accessory CL with GnRH or hCG greatly enhanced the reproductive performance of recipient cows with low fertility. There was no difference observed in the occurrence of pregnancy loss between GnRH- and hCG-treated cows and controls. This meta-analysis demonstrated the benefit of inducing accessory CLs with hCG or GnRH in ET recipient cows, and highlighted the importance of treatment type, time, parity, and initial fertility in achieving higher pregnancy rates.

## Figures and Tables

**Figure 1 vetsci-10-00309-f001:**
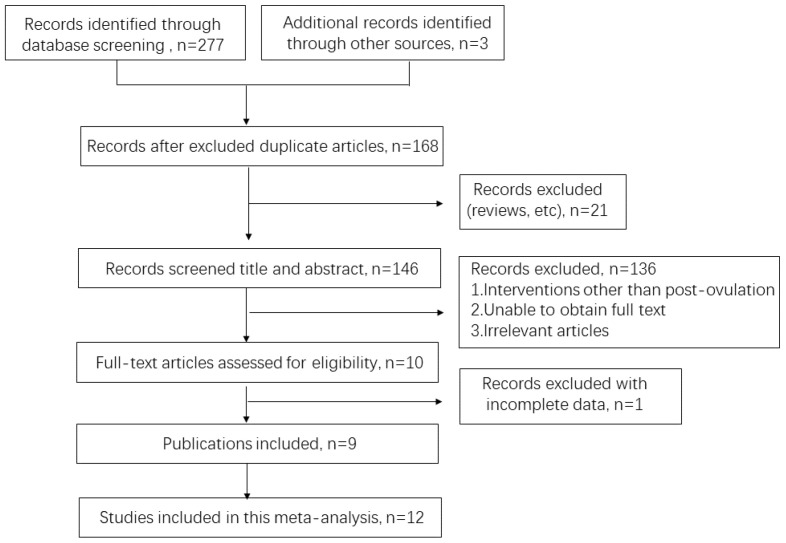
Flow chart for the literature screening.

**Figure 2 vetsci-10-00309-f002:**
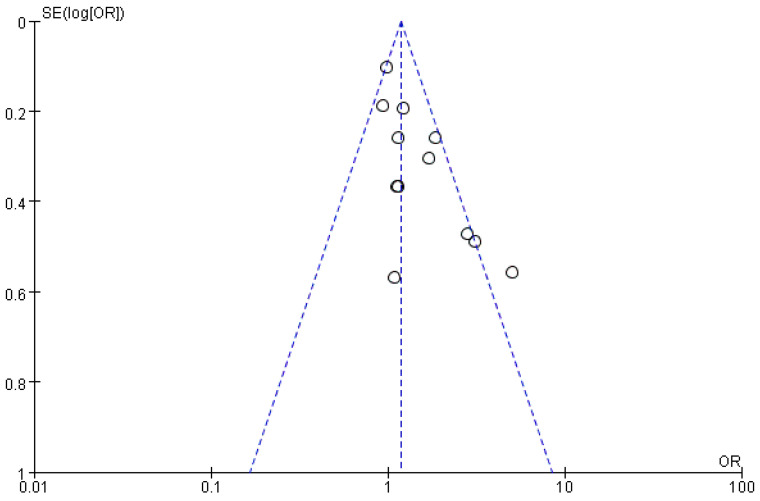
Funnel plot of meta-analysis. Each dot represented a study. The shape of the funnel plots did not reveal any indication of funnel plot asymmetry.

**Figure 3 vetsci-10-00309-f003:**
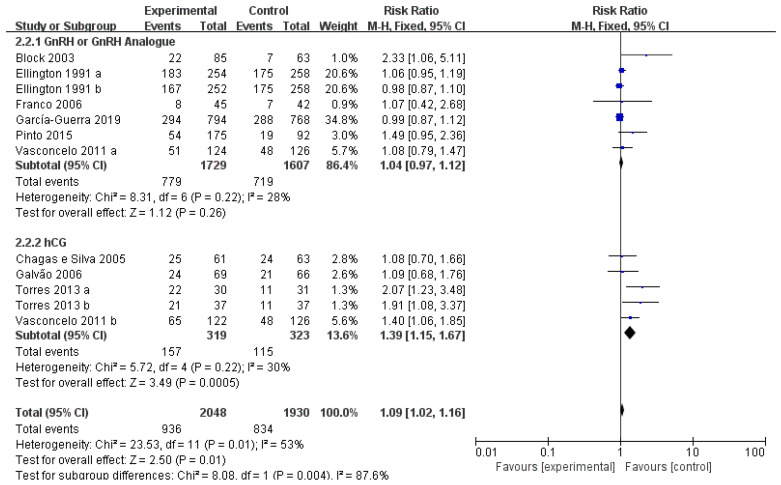
Forest plots: Rist ratio of GnRH and hCG over pregnancy rate of cows after ET. Black boxes refer to the average RR for pregnancy rate improvement with administration regarding every study, with the area proportional to the study contribution to overall analysis. Diamond at bottom denotes pooled RR estimates, and that on right of solid vertical line refers to positive impact of administration on pregnancy rate. Overall RR = 1.09 indicates that administration of GnRH or hCG has favorable impact on fertility.

**Figure 4 vetsci-10-00309-f004:**
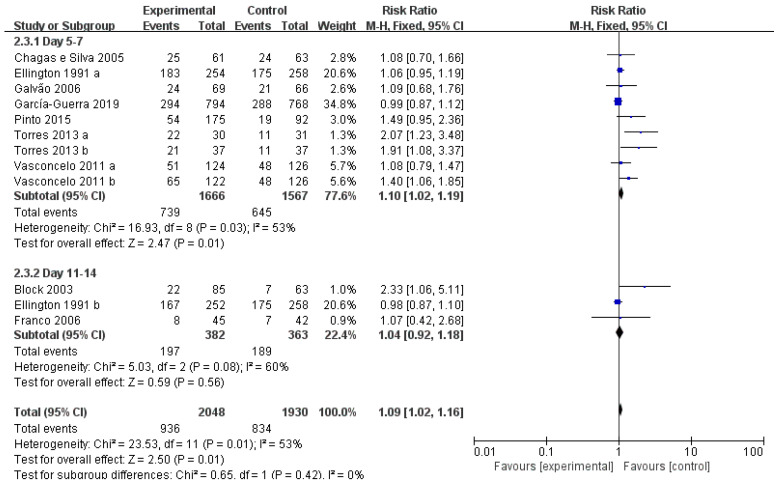
Forest plot: Risk ratio of studies investigating the effect of treatment time on the efficacy of GnRH or hCG treatment on pregnancy rate of cows. Black boxes refer to the average RR for pregnancy rate improvement with administration of every study, with the area proportional to the study contribution to overall analysis. Diamond at bottom denotes pooled RR estimates, and that on right of solid vertical line refers to positive impact of administration on pregnancy rate. As indicated by the results, administration of GnRH or hCG at day 5–7 had a significant benefit on dairy cows.

**Figure 5 vetsci-10-00309-f005:**
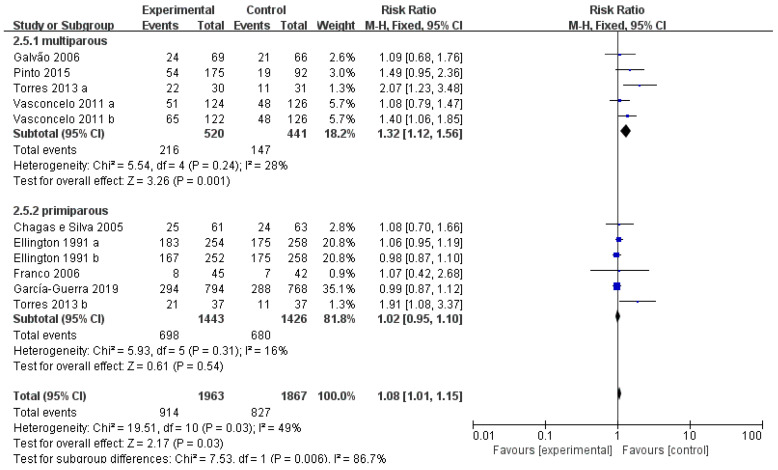
Forest plots: Risk ratio of studies investigating the impact of parity on efficacy of GnRH or hCG on the pregnancy rate of cows after ET. Black boxes refer to the average RR for pregnancy rate improvement with administration of every study, with the area proportional to the study contribution to overall analysis. Diamond at bottom denotes pooled RR estimates, and that on right of solid vertical line refers to positive impact of administration on pregnancy rate. As indicated by the results, administration of GnRH or hCG has significant benefits on multiparous cows.

**Figure 6 vetsci-10-00309-f006:**
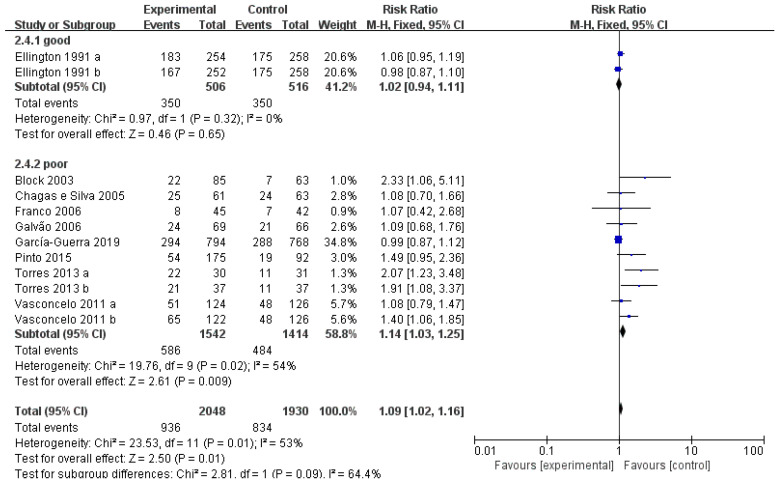
Forest plots: Risk ratio of studies investigating the effect of initial conception rate on efficacy of GnRH or hCG on the pregnancy rate of cows after ET. Black boxes refer to the average RR for pregnancy rate improvement with administration of every study, with the area proportional to the study contribution to overall analysis. Diamond at bottom denotes pooled RR estimates, and that on right of solid vertical line refers to positive impact of administration on pregnancy rate. As indicated by the results, administration of GnRH or hCG have significant benefit on cows with poor pregnancy rates (<40%) but not relatively good fertility (>40%).

**Figure 7 vetsci-10-00309-f007:**
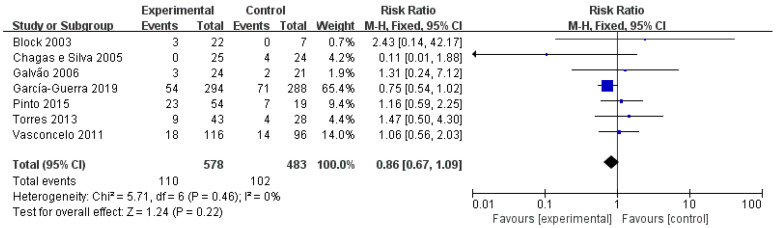
Forest plots: Rist ratio of GnRH and hCG over pregnancy loss rate of cows after ET. Black boxes refer to average RR for pregnancy rate improvement with administration regarding every study, with area proportional to the study contribution to overall analysis. As demonstrated by the results, there is no impact of administration of GnRH or hCG on pregnancy loss (*p* > 0.05).

**Table 1 vetsci-10-00309-t001:** Characteristics of the nine included studies.

Studies	Breed of Recipient	Parity	Treatment Type	Dosage	Treatment Time	Treatment (n)	Control (n)	1st Diagnosis (d)	2nd Diagnosis (d)
Block 2003 [37]	dairy cows	Parous/nulliparous	GnRH	100 µg	11	63	85	53	81
Chagas e Silva 2005 [38]	dairy cows	nulliparous	hCG	1500 IU	7	61	63	28	42
Ellington 1991a [39]	dairy cows	nulliparous	GnRH analogue Buserelin	8 µg	7	254	258	35–60	none
Ellington 1991b [39]	dairy cows	nulliparous	GnRH analogue Buserelin	8 µg	11	252	258	35–60	none
Franco 2006 [40]	dairy cows	nulliparous	GnRH	100 µg	11	45	42	45–53	none
Galvão 2006 [41]	dairy cows	Parous	hCG	3300 IU	5	69	66	28	42
García-Guerra 2019 [42]	dairy cows	nulliparous	GnRH	100 µg	5	794	768	33	60
Pinto 2012 [43]	dairy cows	Parous	GnRH analogue	10 µg Buserelin or 750 µg Deslorelin	7	175	92	37	67
Torres 2013a [44]	dairy cows	Parous/nulliparous	hCG	1500 IU	7	30	31	28	42/63
Torres 2013b [44]	dairy cows			1500 IU	7	37	37	28	42/63
Vasconcelo 2011a [45]	dairy cows	Parous	GnRH/hCG	100 µg	7	124	126	28	41
Vasconcelo 2011b [45]	dairy cows			2500 IU	7	122	126	28	41

## Data Availability

The corresponding author may provide the data used in this research upon request.

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
