# Peer review of "Impact of Accessory Corpus Luteum Induced by Gonadotropin-Releasing Hormone or Human Chorionic Gonadotropin on Pregnancy Rates of Dairy Cattle following Embryo Transfer: A META-Analysis"

_vetsci, 2023, doi:10.3390/vetsci10050309_

Round 1

Reviewer 1 Report (Previous Reviewer 1)

The article highlights the beneficial effect of P4 produced by accessory CL produced in met-estrus following a treatment with gonadotropin and factor that releases it, following ET protocols.

However, the studied articles have limits, they are few in number and unbalanced. They should be balanced between articles with GnRH vs hCG treatment.

There is no balance between the articles that use ET embryos after IVF/IVP vs MOET/IVD

There is no difference between the recipients of embryos, milk cows vs meat cows, there are other conditions, breed, climate....

Most of the recommendations remained unexplained from the initial version.

Author Response

Reviewer 2 Report (Previous Reviewer 2)

the scientific part of the revised version of the paper has been improved. The authors have adequately answered the criticism raised by this reviewer, hence the manuscript is judged as publishable after minor modifications and thorough language corrections. Many parts of the paper are inconceivable, nouns and verbs are misused (example administrated instead of administered, corpora luteum instead of corpora lutea etc)  It is recommended the authors to use either professional services or to have their manuscript revised by a native English speaker.

In terms of the content, the rationale of the study must be clear in the introduction. 

Although it is self-evident for scientists active in the field that the GnRH or HCG was administered in the recipients, this must be clearly mentioned in the last paragraph of the introduction.

Author Response

Reviewer 3 Report (New Reviewer)

The authors made an analysis of embryo transfer publications that use |approach to identify, select, and analyze relevant studies

The analysis suggests that the use of GnRH and hCG can improve pregnancy rates in cows with very poor fertility (<40%), but does not have a significant effect on cows with good fertility. Furthermore, the treatment was more effective in parous lactating dairy cows compared to heifers.

Although the conclusions are not surprising, the study may still have important practical implications. In this case, identifying subgroups of cows that are more likely to benefit from GnRH and hCG treatment may have important implications for the management of reproductive performance in the dairy industry

Author Response

We appreciate the reviewer’s positive evaluation of our work. 

Reviewer 4 Report (New Reviewer)

The objective of the article is interesting and practical. However, I have some concerns regarding the whole manuscript, the methodology and how it is written.

English should be reviewed in the whole article, please, implementing significant changes. There are many sentences that I cannot understand easily or even at all, ant the language is not fluent. I would suggest a thorough revision of the language by a native professional corrector to improve readability.

Abstract: it should be rewritten.

Introduction: It is impossible to make a full review on ET, the objectives, about all causes of infertility of the cows in general… Authors should, please, focus on the review of the item they are working with to let clear why this study was required and is interesting: in fact, progesterone related effects on embryo implantation. Delete any other topics and references. Focus on this, and review this part properly, please.

There are many bibliography on the effects on recipients regarding additional corpus luteum after different treatments. Since decades practitioners work with eCG superovulating lightly recipients to get more embryos... and there are several reviews, and many papers on the topic... please moderate the statement that this was scarcely studied. Why do authors not consider the use of eCG (previously named PMSG) broadly used before? I think, this would enlarge the number of included articles, and would complement very good the study.

M&M: describe better not only the studies included/excluded and the methodology of the Meta analysis, but also the treatments analysed: synchronization, interaction perhaps among age of recipients and sychro programs of recipients? Days of treatment analysed (it is described in results, but nothing said in M&M….

Results: a good description of the articles found and the methodology regarding the meta analysis is made. However, the concrete and practical information is not given. Describe please the treatments analysed. It cannot be describe only, if GnRH or hCG was administered, but when regarding the estrus, after (or not) a synchro program, dosis, even GnRH analogue!!!

Could the result regarding day of treatment and treatment be biased? for example, that those studies administering on day 7 were all or most of them those with hCG? or viceversa? Please clarify.  Similarly regarding results on parity. Authors should analyse possible interaction and baises and clarify these results.

Embryo loss is not the same as pregnancy loss. Please review and correct.

Discussion: a first paragraph should briefly describe the most important results. Aovid to repeat an introduction or description of objectives.

The discussion should be fully rewritten and reviewed. The parts discussing effectos of increasing number of CL in AI cows, if mantaind, should be described in the introduction section (not in the discussion).

Author Response

Reviewer 5 Report (New Reviewer)

The study under review aimed to use meta-analytical methods to evaluate the efficacy of hCG and GnRH in the improvement of pregnancy rates of dairy cows following embryo transfer. This meta-analysis examined 2048 treated cows and 1546 untreated cows extracted from 9 eligible publications and 12 independent studies. Examination showed that administration of GnRH or hCG improved pregnancy rate of cows with poor fertility, while that of cows with good fertility was not affected. Compared to GnRH,  hCG delivered a more consistent results. Improvement in pregnancy rates was greater in cows than in heifers. The evaluated paper contributes new information to the field.

My main concern is to evaluate the effect of GnRH and hCG on embryo loss. Embryonic phase in cattle lasts 42 days (from fertilization to completion of differentiation) and embryonic mortality in cattle is the death of the conceptus before Day 42 of pregnancy (Humblodt, 2002; Bilodeau-Goeseels and Kastelic, 2003) . Then the fetal period begins. Meanwhile, the authors included in their meta-analysis the study in which the first diagnosis of pregnancy was 53 days after estrus (Block et al., 2003) and the second diagnosis of pregnancy in 4 studies ranged from 60 to 81 days after estrus (Block et al., 2003; García-Guerra et al., 2019; Pinto et al., 2012; Torres et al., 2013), thus including the fetal period. Only 3 studies corresponded to the embryonic period criterion. I therefore propose to eliminate the assessment of embryonic losses and rewrite the manuscript accordingly.

Other remarks

I propose to shorten the introduction in the Abstract and add a conclusion at its end.

Table 1 does not indicate the year of publication of Block et al.

Round 2

Reviewer 1 Report (Previous Reviewer 1)

Accept for publication

Author Response

As kindly suggested by the reviewer, we further polished the English language. Abstract, M&M, and discussion were revised to improve the quality of this article. 

We would like to thank the reviewer once more for sparing time on this work.

Reviewer 4 Report (New Reviewer)

Authors have improved notably the manuscript. They have focused on the issue that it is studied, and this makes the readability much better.

However, there are still some issues that can be improved in my opinion: Spelling issues are yet to find in the text. I advise some references to be taken into account.

Authors further consider the use of eCG negligible, although it was one of the first hormones considered for this issue. This should be commented at least in the introduction.

The term "control cows” and the way how authors determined the recipient cows as more or less fertile. Please define this point adequately, such that it is clear from the M&M section onwards....

The discussion section should be reviewed for the grammatical and language. There are some paragraphs difficult to understand yet. Authors insist in including paragraphs in the discussion that are introductory of the topic. Please be shorter and more focused in this section

Authors should be cautious and correct with the use of the term pregnancy loss. The current consensus says that this term refers to pregnancy loss after pregnancy diagnosis, which is only possible after the day 28 of pregnancy, and most often, after the day 32 of pregnancy. In the discussion as well as in the description of the article, there are some parts where this is incorrect or unclear. Please rewrite.

Other /suggestiosn are included in the commented pdf

Author Response

This manuscript is a resubmission of an earlier submission. The following is a list of the peer review reports and author responses from that submission.

Round 1

Reviewer 1 Report

At first glance, the article is good. Analyzing in the field of EmbryoTransfer, I don't think it can be applied as in the field of Artificial Insemination A.I.

The analysis must be done separately for meat/milk cows, primiparous/multiparous, doses, type and time of treatment administration (GnRH/hCG).

The articles seem a bit few and old.

-      The manuscript seems to be composed properly as a review, but it is strikingly similar to other articles in the literature:

Association of pregnancy per artificial insemination with gonadotropin-releasing hormone and human chorionic gonadotropin administered during the luteal phase after artificial insemination in dairy cows: A meta-analysis  doi: 10.3168/jds.2019-16439.

Effects of GnRH and hCG administration during early luteal phase on estrous cycle length, expression of estrus and fertility in lactating dairy cow DOI: 10.1016/j.theriogenology.2021.06.010

The similarity is striking even in legends. The legends are irrelevant and confusing (fig 2).

In the field of reproduction biotechnology, meta-analysis can showed that the use of GnRH and hCG after AI should be focused on cows expected to have low or moderate fertility. Day and dose of treatment have to be considered as well.

In Embryo Transfer, however, I don't think it can be applied because the cows that receive the embryos are chosen through special procedures before starting the protocol. So we don't know which cows will have low fertility, we can use this therapy (GnRH, hCG) for prevention, in this case the meta-analysis has no point.

Progesterone (P4) insufficiency has been reported to be associated with decreased pregnancy after AI/ET. Peripheral P4 concentrations are the net result of secretion (CL) and metabolism (liver). The survival and implantation of the embryo does not occur if P4 is not produced at a sufficient concentration (1 to 2 ng/mL on the day of receiving the embryo (day 6-7 of the sexual cycle).

In order to increase the serum level of P4, these treatments that the authors mention in the title are used in practice, but they have resulted only in certain conditions related to the time/moment of administration, the dose and the existence of the dominant follicle on the ovary. In the absence of precise coordination, your P4 level will not increase, so the result is null. The authors use this treatment in the study but do not explain the method and technique of use (in the material and method) nor the physiological mechanisms (in the discussions)

The manuscript does not respect the guidelines of the author of this journal.

The articles included in the analysis are quite old, only one from 2019. Biotechnology of embryo transfer and embryo production is constantly developing, especially in the last 5-10 years when the IT/science sector exploded. Please visit the archives of the last 5 years of IETS.

The included items are common to both dairy cows and beef cows, they should have been separated. Fertility after ET is different and cannot be compared.

The author's references should be newer, more from the last 5 years. Here are some suggestions:

-      EFFECT OF TREATMENT WITH hCG OR GnRH AT THE TIME OF EMBRYO TRANSFER ON PREGNANCY RATES IN COWS SYNCHRONIZED WITH PROGESTERONE VAGINAL DEVICES, ESTRADIOL BENZOATE, AND eCG

-       

-      Effect of treatment with human chorionic gonadotropin 7 days after artificial insemination or at the time of embryo transfer on reproductive outcomes in nulliparous Holstein heifers.

 DOI:https://doi.org/10.3168/jds.2018-15588

-      Effects of GnRH and hCG administration during early luteal phase on estrous cycle length, expression of estrus and fertility in lactating dairy cowsDOI: 10.1016/j.theriogenology.2021.06.010

- Accessory corpus luteum induced by human chorionic gonadotropin on day 7 or days 7 and 13 of the estrous cycle affected follicular and luteal dynamics and luteolysis in lactating Holstein cows. DOI: 10.3168/jds.2021-20619

- Non-steroidal anti-inflammatory drugs at embryo transfer on pregnancy rates in cows: A meta-analysis. DOI: 10.1016/j.theriogenology.2021.04.010

- https://doi.org/10.1071/RDv35n2Ab129

- https://doi.org/10.1071/RDv35n2Ab189

-  https://doi.org/10.1071/RDv28n2Ab110

 - https://doi.org/10.1071/RDv22n1Ab178

- https://doi.org/10.1071/RD07089

- DOI: 10.1016/j.theriogenology.2019.09.018

Reviewer 2 Report

In this meta-analysis the authors tried to evaluate the effects of GnRH or hCG priming of recipients on pregnancy rate after ET in cattle. Having analyzed the results of a limited number of published papers, the authors concluded that gonadotrophin treatment can improve conception rates only in low fertility cows, and it has neutral effects on embryo survival. This paper carries very limited-if any- novelty,  and -to this reviewer’s opinion- it suffers serious scientific flaws as listed below.

1.     The number of analyzed studies is very small (in table one 15 studies are listed, but the actual number is only 13).

2.     Data and fertility records from beef and dairy cows are mixed. This is inappropriate as the reproductive physiology in the two types of cows differs due to lactation, energy demands, etc. As production type, breed, and age are determining factors for fertility, it appears that the evaluation of some treatments is based on an extremely low number of studies, such as beef hCG only one study. 

3.     The conclusion that low fertility can be improved by gonadotrophin stimulation is rather arbitrary. When condition driving low fertility are not examined and evaluated, it is risky to come in such conclusions. In fact, it appears that authors have not carefully examined the papers used for their study. For example: in the study by Calvao et al, fresh embryos were used, under heat stress conditions that wanes fertility; Nishigan and co-workers used frozen embryos, which give lower conception rates than the fresh ones; Pinto et al used in vitro produced embryos in tropical climate; finally in the study of Torres et al semi embryos were used which by definition result in low conception rates. Hence, the low or high fertility has to be defined under the particular experimental conditions, which in the present study were not taken into account.  

4.     In the discussion the authors comment on P4 supplementation, but in no study progesterone supplementation was carried out, while the results on the efficiency of gonadotrophins to stimulate progesterone secretion is not in concert among studies

5.     The paper needs extensive language improvement

For this reasons, this reviewer believes that the paper should not be given priority for publication in Vet Sciences